# Screening for Preeclampsia and Fetal Growth Restriction in the First Trimester in Women without Chronic Hypertension

**DOI:** 10.3390/jcm12175582

**Published:** 2023-08-27

**Authors:** Piotr Tousty, Magda Fraszczyk-Tousty, Anna Golara, Adrianna Zahorowska, Michał Sławiński, Sylwia Dzidek, Hanna Jasiak-Jóźwik, Magda Nawceniak-Balczerska, Agnieszka Kordek, Ewa Kwiatkowska, Aneta Cymbaluk-Płoska, Andrzej Torbé, Sebastian Kwiatkowski

**Affiliations:** 1Department of Gynecology and Obstetrics, Pomeranian Medical University, 70-111 Szczecin, Poland; 2Department of Neonatology and Neonatal Intensive Care, Pomeranian Medical University, 70-111 Szczecin, Poland; 3Department of Laboratory Diagnostics, Public Clinical Hospital No. 2, 70-111 Szczecin, Poland; 4Department of Nephrology, Transplantology and Internal Medicine, Pomeranian Medical University, 70-111 Szczecin, Poland; 5Department of Reconstructive Surgery and Gynecological Oncology, Pomeranian Medical University, 70-111 Szczecin, Poland

**Keywords:** preeclampsia, fetal growth restriction, screening, first trimester, aspirin

## Abstract

Background: Nowadays, it is possible to identify a group at increased risk of preeclampsia (PE) and fetal growth restriction (FGR) using the principles of the Fetal Medicine Foundation (FMF). It has been established for several years that acetylsalicylic acid (ASA) reduces the incidence of PE and FGR in high-risk populations. This study aimed to evaluate the implementation of ASA use after the first-trimester screening in a Polish population without chronic hypertension, as well as its impact on perinatal complications. Material and methods: A total of 874 patients were enrolled in the study during the first-trimester ultrasound examination. The risk of PE and FGR was assessed according to the FMF guidelines, which include the maternal history, mean arterial pressure (MAP), uterine artery pulsatility index (UtPI), pregnancy-associated plasma protein A (PAPP-A) and placental growth factor (PLGF). Among patients with a risk higher than >1:100, ASA was administered at a dose of 150 mg. Perinatal outcomes were assessed among the different groups. Results: When comparing women in the high-risk group with those in the low-risk group, a statistically significantly higher risk of pregnancy complications was observed in the high-risk group. These complications included pregnancy-induced hypertension (PIH) (OR 3.6 (1.9–7)), any PE (OR 7.8 (3–20)), late-onset PE (OR 8.5 (3.3–22.4)), FGR or small for gestational age (SGA) (OR 4.8 (2.5–9.2)), and gestational diabetes mellitus type 1 (GDM1) (OR 2.4 (1.4–4.2)). The pregnancies in the high-risk group were more likely to end with a cesarean section (OR 1.9 (1.2–3.1)), while the newborns had significantly lower weights (<10 pc (OR 2.9 (1.2–6.9)), <3 pc (OR 10.2 (2.5–41.7))). Conclusions: The first-trimester screening test for PE and FGR is a necessary and effective tool in identifying high-risk pregnancies. ASA prophylaxis among high-risk patients may have the most beneficial effect. Furthermore, this screening tool may significantly reduce the incidence of early-onset PE (eo-PE).

## 1. Introduction

Preeclampsia (PE) and fetal growth restriction (FGR) are significant causes of maternal and fetal mortality worldwide, leading to iatrogenic preterm labor and prolonged hospitalizations for mothers and newborns [1,2].

Until now, groups at risk of these disorders occurring in the first trimester of pregnancy have been identified based on the maternal history of illnesses and previous pregnancies [3,4]. However, in recent years, the Fetal Medicine Foundation (FMF) has demonstrated that these disorders can be predicted using additional factors. This comprehensive assessment includes biochemical indicators such as the placental growth factor (PLGF) and pregnancy-associated plasma protein A (PAPP-A), biophysical markers such as the mean arterial pressure (MAP) and uterine artery pulsatility index (UtA-PI), and the maternal history. Together, these factors are highly effective predictors of PE, with a detection rate (DR) of 90% for its early-onset variety (eo-PE), 75% for preterm PE, and 42% for term PE with a false positive rate (FPR) of 10%. However, the algorithm’s effectiveness in diagnosing FGR is lower, achieving a DR of approximately 50% [5,6,7,8,9,10,11].

Currently, there is no available treatment to prolong pregnancy in confirmed PE cases, and the only effective therapeutic option is to terminate the pregnancy. However, the use of acetylsalicylic acid (ASA) in women at an increased risk of developing PE in the first trimester has been shown to reduce the incidence of PE before 37 weeks’ gestation (wkGA) by 62% [12]. Furthermore, if women with chronic hypertension and those who received less than 90% of the recommended doses are excluded from the study, the risk reduction would be as high as 95%. Unfortunately, the same study found no significant reduction in the incidence of preterm PE among women with chronic hypertension in the aspirin-taking group (5/49) compared to the placebo (5/61) (OR 1.29, 95% CI 0.33–5.12) [13]. ASA has also been found to be useful in cases of increased risk of small for gestational age (SGA), where it has been shown to reduce the incidence of SGA before 37 wkGA by approx. 40–44%. However, this reduction does not extend to the incidence of SGA after the completion of the 37th wkGA [14]. The current recommendations from the International Federation of Gynecology and Obstetrics (FIGO) suggest the use of ASA in high-risk patients starting before 16 wkGA and continuing until 36 wkGA [15]. The primary objective of this study was to assess the effectiveness of the PE and FGR screening test, according to the FMF, followed by administering ASA to a high-risk group of Polish women without chronic hypertension. It is believed that this group of women may derive the greatest benefit from taking ASA. The secondary goal was to compare the perinatal outcomes between groups based on whether the woman was classified in the ASA-taking group and whether PE or FGR were present. To the best of our knowledge, there have been no previous evaluations of ASA use in women at high risk of developing PE and FGR in the Polish population.

## 2. Patients and Methods

This prospective study, conducted from 2019 to 2022, included 908 Caucasian women with healthy singleton pregnancies who were examined in the Pomeranian Medical University’s Second Autonomous Public Clinical Hospital, in the Department of Obstetrics and Gynecology. Patients with chronic hypertension were excluded, resulting in a final enrollment of 874 patients. A first-trimester screening test was performed in each patient to assess aneuploidy, fetal defects, and the risk of developing PE and FGR. The study was conducted following the FMF principles. The Polish healthcare system features a publicly funded prenatal screening program for women aged 35+, who accounted for a significant percentage of the study population. Basic anthropometric measurements were taken, medical histories were obtained, the arterial pressure was measured twice in each arm, and a transabdominal probe was used to determine the UtA-PI. Subsequently, blood samples were collected from each patient for PAPP-A and PlGF concentration measurements, using the Cobas e 801 (Roche Diagnostics, Warsaw, Poland) analyzer. Each patient was then assessed for the risk of eo-PE and FGR based on the FMF algorithms (FMF—2012 software, version 2.8.1). The algorithm for the eo-PE risk assessment consists of a comprehensive assessment of maternal characteristics together with UtPI, MAP, UtPI, and PLGF, with or without PAPP-A, and it is currently based on a paper from 2018 [5]. The authors defined eo-PE according to the International Society for the Study of Hypertension in Pregnancy (ISSHP) criteria [16]. The algorithm for assessing FGR also consists of evaluating the same parameters used to evaluate eo-PE, but it is based on a 2010 paper, where other parameters currently not used in predictions were evaluated (for example, placental protein 13 (PP13) and A Disintegrin and Metalloprotease (ADAM12)). For FGR, the authors used the definition of a birth weight below the fifth percentile [11]. Patients at a high risk (>1:100) of developing eo-PE or FGR were advised to take doses of 150 mg of ASA until 36 wkGA. Perinatal outcomes, such as pregnancy-induced hypertension (PIH), gestational diabetes mellitus (GDM), FGR (in accordance with the Delphi criteria—see Table 1) [17], an SGA diagnosis (estimated fetal weight (EFW) or a fetal abdominal circumference (AC) between the 3rd and 10th percentiles (pc) without any features of FGR), and the presence of PE, were assessed. 

For PE, the criterion used was as defined by the ISSHP. PE was diagnosed if the following criteria were met after 20 wkGA: systolic blood pressure ≥ 140 mm Hg or diastolic blood pressure ≥ 90 mm Hg, along with proteinuria, defined as daily protein loss > 300 mg (or protein:creatinine ratio > 30 mg/mmol). If no proteinuria was found, then at least one of the following criteria had to be satisfied:Hematological disorders (thrombocytopenia, DIC, hemolysis).Serum creatinine content > 1.1 mg/dL or a 2-fold increase in its baseline level where no other kidney disease is observed.Increased serum liver enzymes ≥ 2 times the upper limit of the standard or severe right upper quadrant or epigastric pain.Neurological signs or visual impairment.Pulmonary edema.Intrauterine growth restriction [16].

For each newborn, the following information was assessed: birth week, sex, delivery method, 5-min Apgar score, and basic anthropometric measurements such as the neonatal birth weight. Fenton growth charts (www.ucalgary.ca/fenton accessed on 17 July 2023) were used to determine the birth weight percentiles. The flowchart of the study is shown in Figure 1.

The study was conducted in compliance with the Declaration of Helsinki and received approval from the Institutional Review Board of the Pomeranian Medical University in Szczecin (KB-0012/122/12 of 29 October 2012). Table 2 presents the essential characteristics of the study group, including anthropometric measurements, medical histories of comorbidities, obstetric history, family history, and addictions.

## 3. Statistical Analysis

Data from the study were subjected to statistical analysis. Quantitative data were analyzed using non-parametric Mann–Whitney U tests, while qualitative data were analyzed using either the chi-squared test or Fisher’s exact test. Multivariate logistic regression was performed to calculate the area under the curve (AUC) and odds ratio (OR) for selected parameters. The analysis was conducted using the Statistica software (version 13, StatSoft, Kraków, Poland).

## 4. Results

Table 3 provides an overview of the differences between patients categorized as being at either high or low risk of PE (left-hand side) or FGR (right-hand side) during the first trimester. A high risk of PE was found in 35 of 874 patients (4%). This group included 4 of 19 patients who developed any form of PE (21%) and 6 of 51 patients who were diagnosed with FGR or SGA (11.7%). Patients at high risk for PE demonstrated a statistically significant association with nulliparity (OR 2.4 (1.2–5)). In terms of perinatal outcomes, patients at high risk for PE exhibited a higher likelihood of developing PIH (OR 3.8 (1.6–9.1)), all PE (OR 7.1 (2.2–22.6)), lo-PE (OR 7.6 (2.4–24.4)), and FGR or SGA (OR 3.7 (1.4–9.2)). Additionally, these pregnancies were more frequently concluded with a cesarean section (OR 2.1 (0.98–4.4)), although this result approached statistical significance. However, the high-risk PE group did not show statistically significant differences in terms of maternal age, maternal weight, the development of GDM, pre-pregnancy diabetes, stillbirths, smoking, eo-PE, preterm births, the birth status of the newborn assessed by the Apgar scale, the sex of the newborn, or birth weight (*p* > 0.05). 

In contrast, a high risk of FGR was found in 74 of 874 patients (8.4%). This group included 8 of 19 patients who developed any form of PE (42.1%) and 15 of 51 patients who were diagnosed with FGR or SGA (29.4%). Patients at high risk of FGR during the first trimester were statistically significantly more likely to be nulliparous (OR 2 (1.3–3.3)) and smokers (OR 4.4 (2.1–9.3)). Concerning perinatal outcomes, patients at high risk for FGR demonstrated a higher likelihood of developing GDM1 (OR 3.1 (1.8–5.2)), PIH (OR 4.4 (2.3–8.3)), all PE (OR 8.7 (3.4–22.4)), lo-PE (OR 9.6 (3.7–25.1)), and FGR or SGA (OR 5.4 (2.8–10.4)). Additionally, these pregnancies were more likely to result in a cesarean delivery (OR 1.8 (1.1–2.9)), and the neonatal birth weight was more likely to be <10th percentile (OR 3.2 (1.3–7.7)) and <3rd percentile (OR 11.3 (2.8–46.3)). No statistical significance was found for maternal age, maternal weight, pre-pregnancy diabetes, GDM2, stillbirth, eo-PE, preterm birth, newborn sex, newborn status as assessed by the Apgar scale, or newborn sex among patients at high risk for FGR.

Table 4 presents the differences observed between patients diagnosed with or without PE (left-hand side) and diagnosed with or without FGR or SGA (right-hand side). In the whole group, 19 cases of PE (2.1%) were diagnosed. Patients diagnosed with PE demonstrated a higher likelihood of having FGR or SGA (OR 8.3 (3–22.9)), with their pregnancies being more frequently concluded via cesarean section (OR 3.1 (1.01–9.3)). Furthermore, their newborns were more likely to have a birth weight < 3rd percentile (OR 16.6 (3.1–88.3)). Among patients diagnosed with PE, statistically significantly higher values were observed for MoM UtPI (OR 8.5 (2.4–30.5)) and MoM MAP (OR 32.4 (14.4–55.3)) in the first trimester, while MoM PLGF was significantly lower (OR 0.2 (0.03–0.9)). However, no statistical significance was found for maternal age, maternal weight, nulliparity, pre-pregnancy diabetes, GDM, stillbirth, preterm birth, newborn status as assessed by the Apgar scale, newborn sex, birth weight < 10 pc, or MoM PAPP-A among patients diagnosed with PE. 

There were 51 cases of FGR or SGA (5.8%) in the entire study group. Significantly more patients diagnosed with FGR or SGA were underweight (OR 3.3 (1.2–9.1)) and nulliparous (OR 2.9 (1.6–5.3)). In this group, the incidence of all PE (OR 8.3 (3–22.9)), lo PE (OR 9 (3.2–25.1)), and preterm birth (OR 2.5 (1.1–5.7)) was significantly higher. Regarding newborns, neonatal birth weight was more often <10th percentile (OR 51.4 (22.8–116.5)) and <3rd percentile (OR 17.4 (4.2–71.7)). Patients diagnosed with FGR or SGA exhibited statistically significantly higher MoM UtPI values (OR 2.6 (1.1–6.4)) and lower MoM PLGF values (OR 0.24 (0.1–0.7)) in the first trimester. However, among the patients diagnosed with FGR or SGA, no statistical significance was found for age, normal maternal weight, overweight, obesity, pre-pregnancy diabetes, GDM, smoking, stillbirth, PIH, eo-PE, incidence of cesarian delivery, newborn sex, newborn status as assessed by the Apgar scale, MoM PAPP-A, and MoM MAP values in the first trimester. 

Table 5 summarizes the differences between patients at high or low risk of PE or/and FGR in the first trimester. A high risk of FGR and/or PE was found in 81 of 874 patients (9%). This group included 8 of 19 patients who developed any form of PE (42.1%) and 15 of 51 patients who were diagnosed with FGR or SGA (29.4%). Patients in the high-risk group were significantly more likely to be nulliparous (OR 1.9 (1.2–3)) and smokers (OR 3.9 (1.9–8.2)). In terms of perinatal outcomes, the high-risk group had a higher incidence of gestational diabetes mellitus type 1 (GDM1) (OR 2.4 (1.4–4.2)), pregnancy-induced hypertension (PIH) (OR 3.6, (1.9–7)), all types of PE (OR 7.8 (3–20)), late-onset PE (lo-PE) (OR 8.5 (3.3–22.4)), and FGR or SGA (OR 4.8 (2.5–9.2)). Furthermore, pregnancies in the high-risk group were more likely to result in a cesarean delivery (OR 1.9 (1.2–3.1)). Neonates born to high-risk patients had a higher likelihood of being <10 percentile (OR 2.9 (1.2–6.9)) for birth weight and <3 percentile (OR 10.2 (2.5–41.7)). No statistical significance was found for maternal age, maternal weight, pre-pregnancy diabetes, GDM2, eo-PE, preterm births, stillbirth, <7 Apgar score, or newborn sex. For correlations between first-trimester biochemical and biophysical parameters and the birth weight and birth week in all of the discussed groups, please refer to Appendix A in the Appendix A.

Table 6 presents the DR for screening for all forms of PE, as well as FGR or SGA, in a Polish population without chronic hypertension, followed by the implementation of ASA in the high-risk group. For all forms of PE, the DR was 48% and 61% at an FPR of 5% and 10%, respectively, with an area under the curve (AUC) of 0.85 (0.81–0.89 95%CI). Regarding FGR and SGA, the DR was 20% and 24% at an FPR of 5% and 10%, respectively, with an AUC of 0.70 (0.67–0.73 95%CI). Figure 2 shows receiver operating characteristic (ROC) curves for the relevant parameters.

## 5. Discussion

Our study is the first in Poland to evaluate the efficacy of implementing ASA in pregnancies without chronic hypertension at high risk of PE and FGR, according to the screening principles published after the ASPRE study. The ASPRE study showed the advantages of ASA use in the general population, resulting in a 62% reduction in the incidence of preterm PE and up to an 82% reduction in early-onset (eo-PE) cases, although the latter result bordered on statistical significance [12]. A secondary analysis of the ASPRE study indicated that, if women with chronic hypertension were excluded from the study, consistent ASA use (>90% of the doses) could potentially achieve a 95% reduction in PE incidence [13,18]. 

In our study, none of the women classified as high-risk for eo-PE developed this form of PE. This finding may be attributed to the delayed diagnosis of PE in the later weeks of pregnancy due to the effects of ASA. eo-PE is known to be associated with the abnormal remodeling of spiral arteries, inflammation, and subsequent vascular endothelial damage [19,20,21]. It is speculated that implementing ASA before 16 wkGA in high-risk pregnancies for placental pathologies such as PE or FGR promotes normal spiral artery remodeling and stabilizes the vascular endothelium. As a result, it prevents the development of early-onset forms of PE or FGR or postpones their diagnosis in favor of late-onset forms. This shift in diagnosis leads to significantly improved perinatal outcomes by reducing fetal and maternal morbidity and mortality [22,23].

However, these beneficial effects of ASA are not observed in pregnancies with chronic hypertension. This discrepancy may be due to pre-existing vascular endothelial dysfunction and an ongoing inflammation, which make the development of PE likely even with less severe impairment of spiral vascular remodeling, exacerbating the already existing vascular damage [24].

Our study results demonstrated that the DR for all the forms of PE in our population was 61% at an FPR of 10%, even considering the use of acetylsalicylic acid (ASA), which could potentially affect the DR. To date, the algorithm proposed by the FMF, which incorporates a multivariate analysis including maternal characteristics and history and biochemical and biophysical measurements, is considered the best method for PE detection [5]. It is important to note that the DRs assumed by the FMF algorithm may vary depending on the population in which it is implemented. Previous studies have reported DRs ranging from 41% to 57% at an FPR of 10% when the FMF first-trimester screening test is performed for all forms of PE. However, the DR differs when diagnosing preterm PE or eo-PE, with the same algorithm achieving much higher DR values of up to 90% at an FPR of 10% [25,26,27,28]. Despite this high DR, there is still debate around the world regarding the method for PE screening in the first trimester, as well as the recommended dose of ASA. Scientific societies do not present a unified statement, but, after the ASPRE publication, many countries have changed their recommendations to the approach proposed by the FMF [29]. Our study shows that we still do not have a perfect method for predicting the occurrence of all forms of PE, especially those with a late onset, and many occur in low-risk patients.

When authors compare the FMF algorithm with those proposed by the American College of Obstetricians and Gynecologists (ACOG) and the National Institute for Health and Care Excellence (NICE), the FMF algorithm appears to be the most effective at a relatively low FPR. Following the NICE recommendations, we can detect 41% of preterm PE cases and 34% of term PE cases at an FPR of 10%. On the other hand, according to the new ACOG recommendations, the DR is much higher, reaching up to 90%. However, in the latter case, the FPR can be as high as 60% or more, which may lead to the low acceptance of ASA use among this group of patients and potentially reduce compliance with the recommended treatment [30,31,32]. In our study, we did not present DRs for these forms of PE as there were no eo-PE cases in the group taking ASA. Nonetheless, our results demonstrated that the first-trimester screening test for PE allowed for the identification of the high-risk pregnancy group. A positive test result for PE was associated with a more than seven-fold increase in the risk of developing PE in this group, and up to one in five patients would develop pregnancy complications such as PIH or FGR or be diagnosed with fetal SGA. 

Consequently, our study suggests that we are making progress in detecting and preventing PE, particularly in its early-onset form. However, the prediction of FGR or SGA is a slightly different challenge. The algorithm proposed by the FMF demonstrates a lower DR of 21–44% for term SGA and 46–55% for detecting preterm SGA [14,33]. In our study, we did not achieve satisfactory results in terms of detecting SGA or FGR, with a DR of only 24% at an assumed FPR of 10%. Given these findings, the question arises as to whether we can prevent the occurrence of these disorders despite the low percentage of identified higher-risk pregnancies.

ASA comes to our aid; however, the reduction in the incidence of SGA or FGR is not as significant as it is in the case of PE. Studies suggest that, in cases of increased risk identified in the first trimester, there may be a decrease of approximately 40–44% in the preterm form of these disorders. This decrease is mainly attributed to the reduced incidence of preterm PE and eo-PE. However, no such correlations are observed in cases without PE diagnosis or with a lower incidence of term SGA [10,14,22,23,33,34]. In our study, we demonstrated that pregnant women at an increased risk of FGR were significantly more likely to develop pregnancy complications such as PIH, all PE forms, and FGR, or to be diagnosed with SGA. Furthermore, their pregnancies were more likely to conclude with a cesarean section, and newborns were more likely to have a weight of <10 pc and <3 pc.

What should we recommend to a patient at high risk of developing PE or FGR in the first trimester? It is crucial that we actively collaborate with these patients to ensure the consistent and regular intake of ASA. While ASA might not always be effective, it is currently our only option in preventing the occurrence of these disorders. Consistent intake of ASA is the key to success [13,18]. Second, the close monitoring of these high-risk pregnancies is necessary. As our study has demonstrated, the incidence of other pregnancy complications is much higher in this group. Appropriate and prompt diagnosis may help to improve perinatal outcomes by reducing fetal morbidity and mortality [28]. 

In Poland, the main current focus of the first-trimester screening test is the detection of structural abnormalities and chromosomal abnormalities through ultrasounds and blood sampling for PAPP-A and Beta human chorionic gonadotropin (BHCG). However, not all women are eligible for reimbursement of the test costs, and not all sonographers are certified to identify risks related to PE and FGR. As our study showed, expanding the first-trimester screening test to include additional measurements not only facilitated the implementation of ASA prophylaxis in pregnancies at higher risk of these disorders but also enabled the identification of the high-risk pregnancy group, thus enabling appropriate management. 

## 6. Strength and Limitations

This paper’s strength lies in the inclusion of a large number of women over the age of 35, who are already at higher risk of pregnancy complications due to their age. Another strength is the exploration of screening tests in Poland following the ASPRE trail and the identification of the high-risk group, which has not been validated in the country so far.

As for weaknesses, it should be noted that the study group lacked cases of eo-PE, preventing the determination of DR and AUC for this complication. This may be attributed to the significant reduction in the risk of eo-PE in high-risk populations without chronic hypertension who have received ASA. The use of ASA in our study can be considered controversial, as it has impacted the obtained results. A comparison between high-risk groups with and without ASA administration would be desirable. However, conducting such a study presents ethical challenges. In the present work, we were more interested in showing how screening in the first trimester can help isolate pregnancies that are at the highest risk of perinatal complications. It is also important to mention the potential for errors in the diagnosis of FGR or SGA, especially in the 3–10 pc range. Furthermore, the differentiation between elective and emergency cesarean sections was not addressed, and the monitoring of ASA adherence by the patients was not included, which could have enhanced the value of this study. 

## 7. Conclusions

Our results show the importance and effectiveness of the first-trimester screening test for PE and FGR, particularly in high-risk pregnancies where ASA prophylaxis may have the most beneficial effect. Moreover, the implementation of ASA prophylaxis in pregnancies without chronic hypertension may be especially important in reducing the incidence of eo-PE, as suggested by the absence of such a complication in our high-risk population.

Screening for PE and FGR additionally shows that, even in the absence of an ASA effect, we isolated high-risk pregnancies, meaning that the patients may then receive better perinatal care. However, it should be noted that studies involving a greater number of patients would be necessary to confirm this finding in the Polish population.

## Figures and Tables

**Figure 1 jcm-12-05582-f001:**
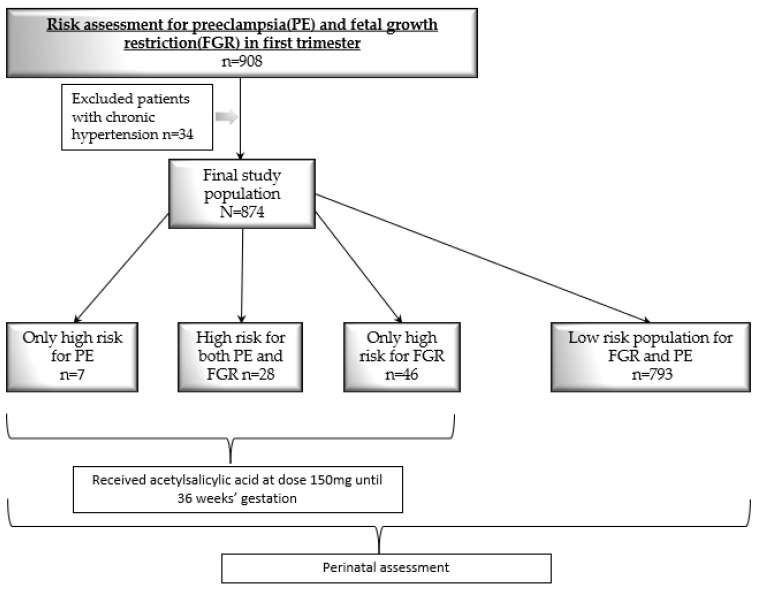
Study flow chart.

**Figure 2 jcm-12-05582-f002:**
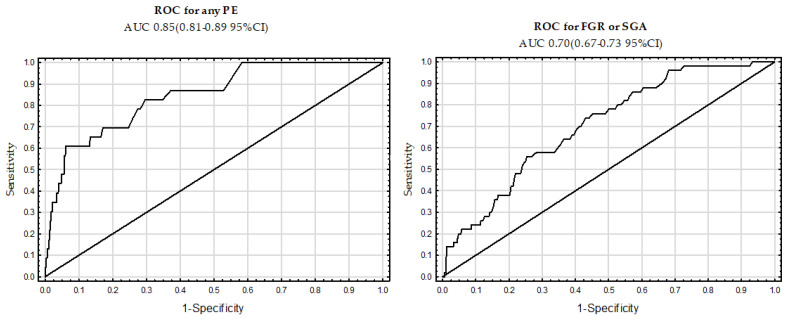
Receiver operating characteristic (ROC) curves for the any form of preeclampsia and fetal growth restriction or small for gestational age.

**Table 1 jcm-12-05582-t001:** Definition of FGR in accordance with the Delphi criteria.

Early FGR: GA < 32 weeks, in the absence of congenital anomalies	Late FGR: GA ≥ 32 weeks, in the absence of congenital anomalies
AC/EFW < 3rd centile or UA-AEDF	AC/EFW < 3rd centile
Or	Or at least two out of three of the following:
AC/EFW < 10th centile combined with UtA-PI > 95th centile and/or UA-PI > 95th centile	AC/EFW < 10th centile AC/EFW crossing centiles > 2 quartiles on growth centiles * CPR < 5th centile or UA-PI > 95th centile

Note: * Growth centiles are non-customized centiles; AC: fetal abdominal circumference; AEDF: absent end-diastolic flow; CPR: cerebroplacental ratio; EFW: estimated fetal weight; GA, gestational age; PI: pulsatility index; UA: umbilical artery; UtA: uterine artery.

**Table 2 jcm-12-05582-t002:** Characteristics of the study group.

Feature	*n* (%)
**Maternal age and weight**	
**Age > 35 yo**	311 (35.6%)
**Age > 40 yo**	48 (5.5%)
**BMI**	
**Underweight (<18.5)**	31 (3.5%)
**Normal weight (18.5–24.9)**	550 (62.9%)
**Overweight (≥25)**	199 (22.8%)
**Obesity (≥30)**	93 (10.6%)
**Comorbidities and addictions**	
**SLE**	6 (0.7%)
**APS**	9 (1%)
**Diabetes mellitus type 1**	4 (0.5%)
**Smoking**	39 (4.5%)
**Obstetrical history**	
**Parous previous PE**	11 (1.3%)
**Previous FGR or SGA fetuses**	16 (1.8%)
**Family history of PE**	5 (0.6%)
**Nulliparous**	390 (44.6%)
**IVF**	12 (1.4%)

**Note:** APS: antiphospholipid syndrome; **BMI: body mass index**; FGR: fetal growth restriction; IVF: In vitro fertilization; SGA: small for gestational age; **SLE: systemic lupus erythematosus; PE: preeclampsia**; yo: years old.

**Table 3 jcm-12-05582-t003:** Selected midgestational parameters and perinatal outcomes among pregnant patients at high risk of FGR or PE during the first trimester, excluding those with chronic hypertension.

	High Risk for PE *n* (%)	Low Risk for PE *n* (%)	*p*	OR (95%CI)	High Risk for FGR *n* (%)	Low Risk for FGR *n* (%)	*p*	OR (95%CI)
	***n* = 35**	***n* = 839**	-		***n* = 74**	***n* = 800**		
**Maternal characteristics, comorbidities and obstetric history**								
**Age > 35**	12 (34.3%)	299 (35.7%)	0.87	-	31 (41.9%)	280 (35%)	0.24	-
**Age > 40**	1 (2.9%)	47 (5.6%)	0.48	-	3 (4.1%)	45 (5.6%)	0.76	-
**Underweight (BMI < 18.5)**	1 (2.9%)	30 (3.6%)	0.81	-	3 (4.1%)	28 (3.5%)	0.93	-
**Normal weight** **(BMI 18.5–24.9)**	21 (60%)	529 (63.1%)	0.71	-	51 (68.9%)	499 (62.5%)	0.27	-
**Overweight (BMI ≥ 25)**	6 (17.1%)	193 (23%)	0.42	-	10 (13.5%)	189 (23.7%)	0.08	-
**Obesity (BMI ≥ 30)**	7 (20%)	86 (10.3%)	0.07	-	10 (13.5%)	83 (10.4%)	0.41	-
**Nulliparous**	23 (65.7%)	367 (43.7%)	**0.01**	**2.4 (1.2–5)**	45 (60.8%)	345 (43.1%)	**0.003**	**2 (1.3–3.3)**
**Smoking**	0	39 (5.5%)	0.31	-	11 (16.2%)	28 (4.2%)	**<0.001**	**4.4 (2.1–9.3)**
**PGDM**	1 (2.9%)	3 (0.4%)	0.39	-	1 (1.4%)	3 (0.4%)	0.77	-
**Maternal and perinatal outcome**								
**GDM1**	9 (25.7%)	106 (12.6%)	0.03	-	19 (25.7%)	96 (12%)	**<0.001**	**3.1 (1.8–5.2)**
**GDM2**	2 (5.7%)	93 (11.1%)	0.47	-	7 (9.5%)	88 (11%)	0.68	-
**PIH**	7 (20%)	52 (6.2%)	**0.001**	**3.8 (1.6–9.1)**	15 (20.3%)	44 (5.5%)	**<0.001**	**4.4 (2.3–8.3)**
**All PE**	4 (11.4%)	15 (1.8%)	**<0.001**	**7.1 (2.2–22.6)**	8 (10.8%)	11 (1.4%)	**<0.001**	**8.7 (3.4–22.4)**
**eo-PE**	0	1 (0.12%)	-	-	0	1 (0.13%)	-	-
**lo-PE**	4 (11.4%)	14 (1.7%)	**<0.001**	**7.6 (2.4–24.4)**	8 (10.8%)	10 (1.3%)	**<0.001**	**9.6 (3.7–25.1)**
**Cesarean delivery**	25 (71.4%)	458 (54.8%)	0.05	2.1 (0.98–4.4)	50 (67.6%)	433 (54.3%)	**0.03**	**1.8 (1.1–2.9)**
**Preterm birth**	2 (5.7%)	62 (7.4%)	0.71	-	6 (8.1%)	58 (7.3%)	0.79	-
**Newborn outcome**								
**FGR or SGA**	6 (17.1%)	45 (5.4%)	**0.003**	**3.7 (1.4–9.2)**	15 (20.3%)	36 (4.5%)	**<0.001**	**5.4 (2.8–10.4)**
**Stillbirth**	0	3 (0.4%)	0.26	-	1 (1.4%)	2 (0.25%)	0.61	-
**Newborn sex (male)**	20 (57.1%)	454 (54.1%)	0.72	-	42 (56.8%)	432 (54%)	0.65	-
**Apgar score < 7 at 5 min**	1 (2.9%)	15 (1.9%)	0.82	-	2 (2.7%)	15 (1.9%)	0.96	-
**Birth weight < 10 pc**	3 (8.6%)	29 (3.5%)	0.26	-	7 (9.5%)	25 (3.1%)	**0.005**	**3.2 (1.3–7.7)**
**Birth weight < 3 pc**	1 (2.9%)	7 (0.8%)	0.75	-	4 (5.4%)	4 (0.5%)	**<0.001**	**11.3 (2.8–46.3)**

Note: all PE: all preeclampsia types; BMI: body mass index; CI: confidence interval; eo-PE: early-onset preeclampsia; FGR: fetal growth restriction; GDM1: gestational diabetes mellitus type 1; GDM2: gestational diabetes mellitus type 2; lo-PE: late-onset preeclampsia; OR: odds ratio; pc: percentile; PGDM: pregestational diabetes mellitus; PIH: pregnancy-induced hypertension; SGA: small for gestational age.

**Table 4 jcm-12-05582-t004:** Selected midgestational parameters and perinatal outcomes among pregnant patients diagnosed with PE and those diagnosed with FGR or SGA, excluding those with chronic hypertension.

	PE Diagnosis *n* (%)	without PE Diagnosis *n* (%)	*p*	OR (95%CI)	FGR or SGA Diagnosis *n* (%)	without FGR or SGA Diagnosis *n* (%)	*p*	OR (95%CI)
	** *n* ** **= 19**	** *n* ** **= 855**			** *n* ** **= 51**	** *n* ** **= 823**		
**Maternal characteristics, comorbidities and obstetric history**								
**Age > 35**	7 (36.8%)	304 (35.6%)	0.91	-	17 (33.3%)	304 (37%)	0.75	-
**Age > 40**	2 (10.5%)	46 (5.4%)	0.33	-	1 (2%)	47 (5.7%)	0.25	-
**Underweight (BMI < 18.5)**	1 (5.3%)	30 (3.5%)	0.82	-	5 (9.8%)	26 (3.2%)	**0.04**	**3.3 (1.2–9.1)**
**Normal weight (BMI 18.5–24.9)**	13 (68.4%)	537 (62.9%)	0.61	-	35 (68.6%)	515 (62.6%)	0.4	
**Overweight (BMI ≥ 25)**	2 (10.5%)	197 (23.1%)	0.31	-	9 (17.7%)	190 (23.1%)	0.32	
**Obesity** **(BMI ≥ 30)**	3 (15.8%)	90 (10.5%)	0.72	-	2 (3.9%)	92 (11.2%)	0.14	
**Nulliparous**	12 (63.2%)	378 (44.2%)	0.1	-	35 (68.6%)	355 (43.1%)	**<0.001**	**2.9 (1.6–5.3)**
**Smoking**	0	39 (5.4%)	0.66	-	4 (8%)	35 (5.1%)	0.58	-
**PGDM**	0	4 (0.5%)	0.16	-	0	4	-	-
**Maternal and perinatal outcome**								
**GDM1**	2 (10.5%)	113 (13.2%)	0.73	-	9 (17.7%)	106 (12.9%)	0.43	-
**GDM2**	0	95 (11.1%)	0.24	-	3 (5.9%)	92 (11.2%)	0.28	-
**PIH**	-	-	-	-	7 (13.7%)	50 (6.1%)	0.06	-
**All PE**	-	-	-	-	6 (11.8%)	13 (1.6%)	**<0.001**	**8.3 (3–22.9)**
**eo-PE**	-	-	-	-	0	1 (0.1%)	-	-
**lo-PE**	-	-	-	-	6 (11.8%)	12 (1.5%)	**<0.001**	**9 (3.2–25.1)**
**Cesarean delivery**	15 (79%)	469 (55%)	**0.04**	**3.1 (1.01–9.3)**	30 (58.8%)	453 (55.2%)	0.62	-
**Preterm birth**	3 (15.8%)	61 (7.1%)	0.15	-	8 (15.7%)	56 (6.8%)	**0.04**	**2.5 (1.1–5.7)**
**Newborn outcome**								
**FGR or SGA**	6 (31.6%)	45 (5.3%)	**<0.001**	**8.3 (3–22.9)**	-	-	-	-
**Stillbirth**	0	3 (0.4%)	0.8	-	2 (3.9%)	1 (0.1%)	-	-
**Newborn sex (male)**	6 (31.6%)	468 (54.7%)	0.05	2.6 (0.98–7)	23 (45.1%)	451 (54.8%)	0.18	-
**Apgar score <7 at 5 min**	0	17 (2%)	0.82	-	2 (3.9%)	15 (1.8%)	0.6	-
**Birth weight < 10 pc**	2 (10.5%)	30 (3.5%)	0.11	-	21 (41.2%)	11 (1.3%)	**<0.001**	**51.4 (22.8–116)**
**Birth weight < 3 pc**	2 (10.5%)	6 (0.7%)	**<0.001**	**16.6 (3.1–88.3)**	4 (7.8%)	4 (0.5%)	**<0.001**	**17.4 (4.2–71.7)**
**First trimester biochemical or biophysical measurement**								
	Median (min-max)	Median (min-max)	*p*	OR (95%CI)	Median (min-max)	Median (min-max)	*p*	OR (95%CI)
**MoM UtPI**	1.26 (0.6–1.8)	0.98 (0.4–2.3)	**<0.001**	**8.5 (2.4–30.5)**	1.06 (0.7–1.7)	0.98 (0.4–2.3)	**0.03**	**2.6 (1.1–6.4)**
**UtPI**	2.1 (0.9–2.86)	1.5 (0.6–3.8)	**0.002**	**3.5 (1.6–7.8)**	1.8 (1.1–2.6)	1.5 (0.6–3.8)	**0.01**	**2 (1.2–3.5)**
**MoM PAPP-A**	0.87 (0.2–3.1)	0.96 (0.2–4.8)	0.32	-	0.79 (0.2–2.8)	0.96 (0.2–4.8)	0.12	-
**PAPP-A (IU/I)**	2.8 (0.5–12.4)	3.4 (0.5–21.4)	0.53	-	3.3 (0.5–14)	3.4 (0.5–21.4)	0.49	-
**MoM PLGF**	0.78 (0.2–1.63)	0.9 (0.1–3.2)	**0.04**	**0.2 (0.03–0.9)**	0.82 (0.2–1.9)	0.9 (0.1–3.2)	**0.005**	**0.24 (0.1–0.7)**
**PLGF (ng/mL)**	39.4 (12.6–98)	50.4 (11–357)	**0.04**	**0.97 (0.94–0.99)**	46 (11–100)	50.5 (60–357)	**0.01**	**0.98 (0.96–0.99)**
**MoM MAP**	1.12 (0.9–1.4)	1.03 (0.7–1.4)	**<0.001**	**32.4 (14–55.3)**	1.03 (0.8–1.3)	1.03 (0.7–1.4)	0.88	-
**MAP (mm Hg)**	95 (78.3–113)	87 (60–123)	**<0.001**	**1.09 (1.04–1.14)**	85 (72–112)	88 (60–123)	0.27	-

Note: all PE: all preeclampsia types; BMI: body mass index; CI: confidence interval; eo-PE: early –onset preeclampsia; FGR: fetal growth restriction; GDM1: gestational diabetes mellitus type 1; GDM2: gestational diabetes mellitus type 2; lo-PE: late –onset preeclampsia; MoM: multiple of the median; OR: odds ratio; pc: percentile; PAPP-A: Pregnancy Associated Plasma Protein-A; PGDM: pregestational diabetes mellitus; PIH: pregnancy induced hypertension; PLGF: placental growth factor; SGA: small for gestational age; UtPI: uterine artery pulsatility index.

**Table 5 jcm-12-05582-t005:** Selected midgestational parameters and perinatal outcomes among pregnancies at high risk of FGR and/or PE in the first trimester, excluding those with chronic hypertension.

	High Risk for PE or/and FGR *n* (%)	Low Risk for PE and FGR *n* (%)	*p*	OR (95%CI)
	***n* = 81**	***n* = 793**	**-**	
**Maternal characteristics, comorbidities, and obstetrical history**				
**Age > 35**	35 (43.2%)	276 (34.8)	0.13	-
**Age > 40**	3 (3.7%)	45 (5.7%)	0.45	-
**Underweight (BMI < 18.5)**	3 (3.7%)	28 (3.5%)	0.93	-
**Normal weight (BMI 18.5–24.9)**	53 (65.4%)	497 (62.8%)	0.63	-
**Overweight (BMI ≥ 25)**	13 (16.1%)	186 (23.5%)	0.12	-
**Obesity (BMI ≥ 30)**	12 (14.8)	81 (10.2%)	0.2	-
**Nulliparous**	48 (59.3%)	342 (43.1%)	**<0.01**	**1.9 (1.2–3)**
**Smoking**	11 (14.7%)	28 (4.2%)	**<0.001**	**3.9 (1.9–8.2)**
**PGDM**	1 (1.2%)	3 (0.4%)	0.82	-
**Maternal and perinatal outcomes**				
**GDM1**	20 (24.7%)	95 (12%)	**<0.01**	**2.4 (1.4–4.2)**
**GDM2**	7 (8.6%)	88 (11.1%)	0.49	**-**
**PIH**	14 (17.3%)	43 (5.43%)	**0.001**	**3.6 (1.9–7)**
**All PE**	8 (9.9%)	11 (1.4%)	**<0.001**	**7.8 (3–20)**
**eo-PE**	0	1 (0.13%)	0.15	-
**lo-PE**	8 (9.8%)	10 (1.3%)	**<0.001**	**8.5 (3.3–22.4)**
**Cesarean delivery**	56 (69.1%)	427 (54%)	**0.009**	**1.9 (1.2–3.1)**
**Preterm birth**	6 (7.4%)	58 (7.3%)	0.97	-
**Newborn outcome**				
**FGR or SGA**	15 (18.5%)	36 (4.5%)	**<0.001**	**4.8 (2.5–9.2)**
**Stillbirth**	1 (1.2%)	2 (0.25%)	0.15	-
**Newborn sex (male)**	45 (55.5%)	429 (54.1%)	0.8	-
**Apgar score < 7 at 5 min**	2 (2.5%)	15 (1.9%)	0.71	-
**Birth weight < 10 pc**	7 (8.6%)	25 (3.1%)	**0.01**	**2.9 (1.2–6.9)**
**Birth weight < 3 pc**	4 (4.9%)	4 (0.5%)	**<0.001**	**10.2 (2.5–41.7)**

Note: all PE: all preeclampsia types; BMI: body mass index; CI: confidence interval; eo-PE: early-onset preeclampsia; FGR: fetal growth restriction; GDM1: gestational diabetes mellitus type 1; GDM2: gestational diabetes mellitus type 2; lo-PE: late-onset preeclampsia; OR: odds ratio; pc: percentile; PGDM: pregestational diabetes mellitus; PIH: pregnancy-induced hypertension; SGA: small for gestational age.

**Table 6 jcm-12-05582-t006:** Performance of the Fetal Medicine Foundation’s algorithm for the different groups.

	AUC	CI (95%)	Sensitivity for the FPR	
**Variables**			5%	10%
**Any PE**	0.85	(0.81–0.89)	48	61
**FGR or SGA**	0.71	(0.67–0.75)	20	24

Note: AUC: area under the curve; CI: confidence interval; FGR: fetal growth restriction; FPR: false positive ratio; PE: preeclampsia; SGA: small for gestational age.

## Data Availability

The data presented in this study are available upon request from the corresponding author. The data are not publicly available, as not all patients agreed to publicly disclose their data.

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
