# Peer review of "Screening for Preeclampsia and Fetal Growth Restriction in the First Trimester in Women without Chronic Hypertension"

_jcm, 2023, doi:10.3390/jcm12175582_

Round 1

Reviewer 1 Report

The publication is an attempt to claim the ASA as new prophylaxis on risky pregnancies. The publaction is written in good and understandable manner. Data acquired seems solid and paper describes almost thousand of patients, although high risk for disorders had only 81 patients and only 1 from low-risk developed early onset PE group. 

It's hard to evaluate the real beneficial meaning of the early screening testing as the numbers of high risk and low risk are almost 10 fold different. Then the percentages are hard to interpret. FPR seems quite high for this analysis.

PE developed only 10% of high risk group. Altogether PE was developed by 2,17% out of the whole group. Is this accurate this is to be stated and perhaps lightly diluted with the statement. 

Not to be negative, I appreciate the study. It's vast and in overall correct. 

Although, scientific-wise I lack a group of low-risk + ASA. I understand that's it's undoable. But it is clear that from the vast low-risk group plenty developed disorders. It could be visualised that it happened with a table or a schematic in the discussion part. Or at least BOLDED how many were predicted to develop PE or FGR and how many did develop. Like an accuracy score.

Unfortunately, the effect of aspirin is hard to evaluate without proper controls. Furthermore, Aspirin was long time known to be the only one to "treat" PE. But I appreciate progresive study.

For me this work is for publication, with minor addition of accuracy and diluted strong statements of ASA or screening prediction.

So to sum up 

Reviewer 2 Report

The authors emphasize the need and effectiveness of early pregnancy screening, particularly in high-risk pregnancies, where taking ASA as a preventive strategy can have significant advantages in identifying and managing problems, including PE and FGR. Our group of high-risk cases also showed that when ASA is administered as a preventive intervention in pregnancies without persistent hypertension, it significantly reduces the risk of developing early-onset preeclampsia (eo-PE).

Even though the precise effect of ASA is not immediately apparent, this screening method not only aids in identifying pregnancies with a high risk of problems but also leads to better care throughout the postnatal period. However, it's crucial to understand that additional research involving a broader population of patients is necessary to corroborate these results, particularly in the Polish people.

The tables could be clearer to understand. I suggest changing the table style and that the thickness of each line be homogeneous in each table.

The numbers are minimal, and there is a lot of space between rows and columns. I suggest reducing this size, which needs to be used.

Please check the pasive voice when describing a result. 

Reviewer 3 Report

The paper by Tousty et al. explores the accuracy of Fetal Medicine Foundation risk score for PE in a Polish population of around 800 women without chronic hypertension. The paper is overall well written and methods are adequate although this reviewer is concerned about the novelty (FMF score has already been widely used and validated in diverse populations). I have few comments:

- It would be interesting to see tabulated the frequencies of PE familiarity among participants together with some clinical data (e.g. blood pressure values, PI of right and left uterine artery instead of mean value, lab data). In particular it would be interesting to test if adding other variables to FMF risk score increases the accuracy. For example uric acid and serum uric acid to creatinine ratio have shown predictive value in some studies and it would be of great interest to see if in the present study they are able to increase the predictive value of FMF score. As previously said I personally think the present study lacks originality and novelty.

- Please order the acronyms below each Table/Figure in alphabetical order (in particular Table 4).

Round 2

Reviewer 3 Report

The Authors have appropriately addressed my concerns. I still have concerns regarding the originality but the paper itself is suitable for publication.